# A Comprehensive Survey of Contamination Detection Methods in Large Language Models

**Mathieu Ravaut**                                            *mathieuj001@e.ntu.edu.sg*
*Nanyang Technological University*
*Institute of Infocomm Research (I2R), A\*STAR*

**Bosheng Ding**                                              *bosheng001@e.ntu.edu.sg*
*Nanyang Technological University*

**Fangkai Jiao**                                              *fangkai002@e.ntu.edu.sg*
*Nanyang Technological University*
*Institute of Infocomm Research (I2R), A\*STAR*

**Hailin Chen**                                              *hailin001@e.ntu.edu.sg*
*Nanyang Technological University*

**Xingxuan Li**                                              *xingxuan001@e.ntu.edu.sg*
*Nanyang Technological University*

**Ruochen Zhao**                                              *ruochen002@e.ntu.edu.sg*
*Nanyang Technological University*

**Chengwei Qin**                                              *chengwei003@e.ntu.edu.sg*
*Nanyang Technological University*

**Caiming Xiong**                                              *cxiong@salesforce.com*
*Salesforce Research*

**Shafiq Joty**                                              *sjoty@salesforce.com*
*Nanyang Technological University*
*Salesforce Research*

**Reviewed on OpenReview:** *https://openreview.net/forum?id=SxNMjbtdFm*

## Abstract

With the rise of Large Language Models (LLMs) in recent years, abundant new opportunities are emerging, but also new challenges, among which contamination is quickly becoming critical. Business applications and fundraising in Artificial Intelligence (AI) have reached a scale at which a few percentage points gained on popular question-answering benchmarks could translate into dozens of millions of dollars, placing high pressure on model integrity. At the same time, it is becoming harder and harder to keep track of the data that LLMs have seen; if not impossible with closed-source models like GPT-4 and Claude-3 not divulging any information on the training set. As a result, contamination becomes a major issue: LLMs' performance may not be reliable anymore, as the high performance may be at least partly due to their previous exposure to the data. This limitation jeopardizes real capability improvement in the field of NLP, yet, there remains a lack of methods on how to efficiently detect contamination. In this paper, we survey all recent work on contamination detection with LLMs, analyzing their methodologies and use cases to shed light on the appropriate usage of contamination detection methods. Our work calls the NLP research community's attention into systematically taking into account contamination bias in LLM evaluation.

# 1 Introduction

In the rapidly evolving landscape of artificial intelligence (AI), large language models (LLMs) have emerged as a pivotal tool, driving innovation across a wide spectrum of applications, from natural language processing (NLP) and automated content creation (Achiam et al., 2023; Betker et al., 2023) to complex decision-making systems and autonomous agents (Wei et al., 2022; Li et al., 2023a; Yang et al., 2023b; Wu et al., 2023). At their core, these models rely on extensive datasets (Raffel et al., 2020; Gao et al., 2020) to learn about language and the world, and generate responses that are increasingly indistinguishable from human-authored writing (Ouyang et al., 2022a). However, the integrity of these datasets is paramount, as any *contamination* can significantly impair the models' effectiveness, reliability, and generalization. *Contamination* refers to leakage of evaluation data within the training data, an undesirable phenomenon which will inflate the model's performance on the evaluation data.

We identify several primary sources of such contamination that can arise during large-scale data collection and curation. First, unfiltered web scraping to collect pre-training data often introduces data with minimal oversight, increasing the likelihood of inadvertently capturing benchmark or test datasets that are publicly available online. Second, contamination may result from the unintentional inclusion of evaluation materials during preprocessing or aggregation of training corpora, particularly when datasets are not rigorously audited. Finally, the reuse of proprietary or copyrighted material without robust provenance tracking can also lead to inadvertent overlap between training and evaluation sets. These factors underscore the need for meticulous dataset design and transparent documentation practices to ensure the validity of experimental results.

Contamination poses a multifaceted challenge, threatening not only the technical accuracy of LLMs but also their ethical and commercial viability. In high-stakes scenarios where imprecision can have dramatic consequences, such as medical diagnosis, legal advice, or financial services, the repercussions of relying on contaminated data can be profound. Moreover, the allure of leveraging LLM outputs to attract investment underscores a pressing commercial dimension. As businesses increasingly integrate AI-driven insights into their strategic planning and operational decisions, the assurance of data purity becomes intertwined with potential market success and valuation. Lastly, in the current intense race to build the most powerful LLM (Touvron et al., 2023b; Team et al., 2023; Jiang et al., 2024a; Bai et al., 2023; Bi et al., 2024; Young et al., 2024; Gemma Team Google DeepMind, 2024; Chen et al., 2023), the community is struggling to settle on a fixed subset of benchmarks as LLMs performance increases fast (Clark et al., 2018; Zellers et al., 2019; Hendrycks et al., 2020; Lin et al., 2021; Sakaguchi et al., 2021; Cobbe et al., 2021), an issue which is further fueled by underlying contamination. This landscape necessitates a comprehensive survey of contamination detection in LLMs.

## 1.1 Contamination and Contamination Detection Types

Contamination may span only part of a (x, y) data point where x is the input and y the corresponding label. If we consider whether the contaminated data contains only x or both x and y, contamination can be divided into *input contamination* and *input+label contamination*.

Detecting contamination in a LLM is a broad issue that we divide into *open-data* contamination detection and *closed-data* contamination detection depending on the nature of the training set. On the one hand, *open-data* refers to the scenario where the LLM pre-training data is known, enabling direct comparisons with the evaluation dataset. On the other hand, *closed-data* contamination refers to the more and more prevalent use case of an unknown pre-training set. This latter case is more challenging and necessitates an examination of the LLM's behavior on evaluation data points.

Detecting contamination is further complicated by varying level of access to the LLM. With *white-box* access, the LLM and all weights are available, *e.g.,* a local model. In *gray-box* access, model weights are not available to the researcher but output token probability distributions are. Most closed-source setups, *e.g.,* GPT-4, follow a *black-box* setup where only the output text can be accessed from the API. We will review in this study contamination detection methods specifically tailored for each of these levels of model access.

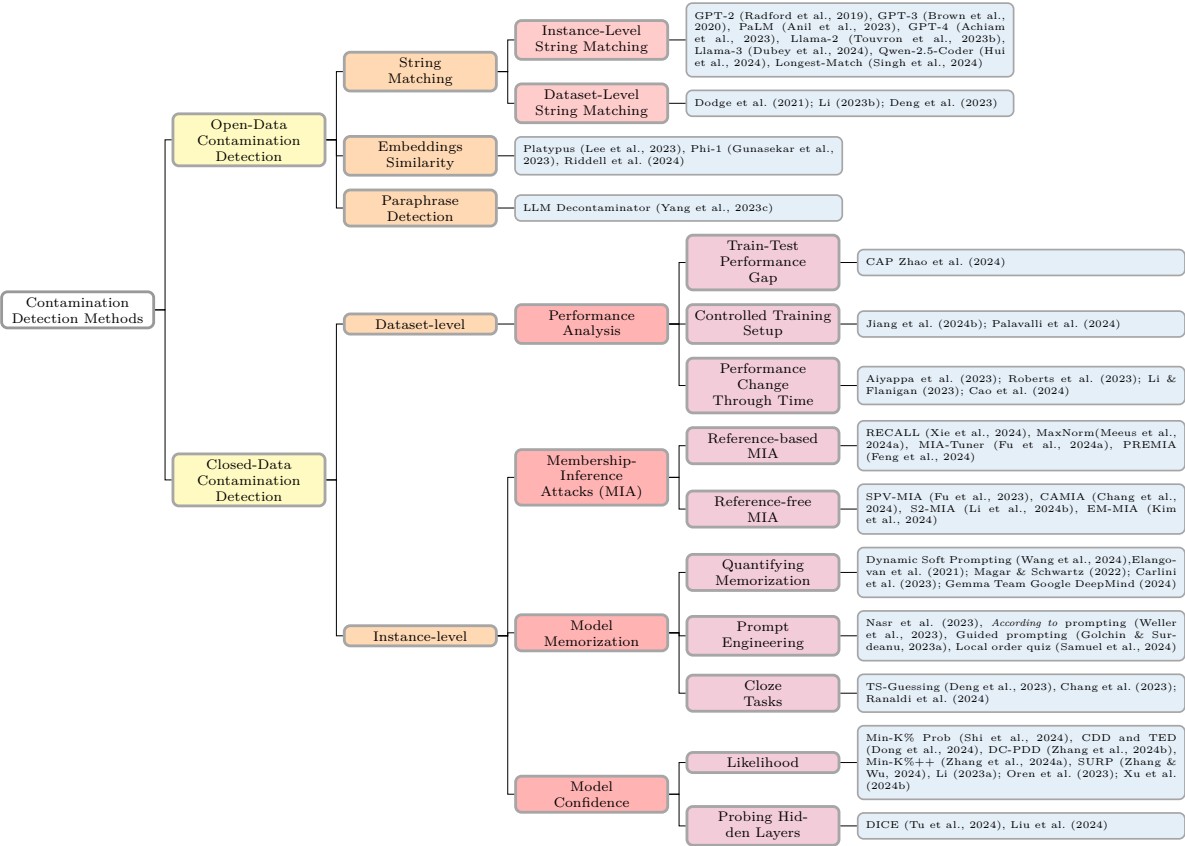

Figure 1: Classification of contamination detection methods reviewed in this paper.

## 1.2 Contributions and Structure of this Survey

To push forward research in contamination detection, we make the following contributions:

- To our knowledge, we are one of the first to thoroughly review the field of contamination detection in LLMs, unrestricted to a specific type of methods or a specific performance aspect. We review all reported contamination detection methods up until early 2025, totaling more than 50 detection techniques and more than 120 relevant papers. We followed an inclusive approached and kept all open-access papers relevant to the problem of contamination detection in LLMs, regardless of publication status. The vast majority of resulting papers which are included in this survey are from 2023-2025.

- We categorize contamination detection into two broad types of use cases vastly differing in the techniques involved, namely *open-data* and *closed-data* contamination detection, and review all existing work in each category. Our taxonomy further classifies detection techniques into finer-grained categories, each following their own assumptions and model requirements.

- Through our detailed classification and review of the types of contamination detection, we not only contribute to the academic and practical understanding of data contamination issues but we also highlight the pressing need for strategies that mitigate these risks.

We refer to Figure 1 for an overview of our proposed classification of contamination detection methods.

The rest of this paper is structured as follows. Section 2 formally defines contamination and associated concepts. Section 3 explores methods and findings related to open-data contamination detection. Section 4 addresses the issue of closed-data contamination detection. Section 5 discusses the current and future challenges in the field, including best practices, and reviews all new datasets proposed to detect contamination.

### 1.3 Comparison to Related Surveys

There already exists surveys on the broad issue of contamination.

Deng et al. (2024) survey contamination in language models, covering the problem from detection, to its impact and methods of remediating it. Compared to their work, our survey solely focuses on detecting contamination, and covers a more exhaustive set of contamination detection techniques.

The work of Palavalli et al. (2024) proposes a taxonomy of the different types of contamination. In Figure 1, our second-level breakdown of closed-data contamination detection into *dataset-level* methods and *instance-level* methods echos their categories. Unlike their survey, we review contamination detection only, and we do so at large, unrestricted to a specific form of contamination such as a particular noising then injection of test data points into the training set.

Unlike the two surveys above, Fu et al. (2024b) focus on contamination detection, like our work. They also propose a taxonomy of detection methods, which has a few similarities with ours (like instance similarity, which we cover within open-data methods), but our classification is more fine-grained (four levels of classification, whereas their classification only has one). The authors review assumptions behind certain detection methods and challenge their true efficiency. Notably, they show that membership-inference attacks (MIAs) perform poorly in detecting pre-training contamination. Our study does not specifically focus on the success of contamination detection methods: we are rather interested in describing in depth the whole landscape of contamination detection, and explain how each method works, including requirements in terms of model access (Table 2) and best suited training stage (Table 3).

## 2 Formal Definitions

Formally, let $M$ be a model (e.g. Llama-3 (Dubey et al., 2024)) and $\mathcal{D}$ be a data distribution on an input space $\mathcal{X}$. We denote by $D_M$ the dataset that $M$ was pre-trained on (e.g, The Pile (Gao et al., 2020)), and $D_{\text{eval}}$ an evaluation dataset of interest (e.g, MMLU (Hendrycks et al., 2020)).

**Definition 1 (Contamination)** *We say that the dataset $D_{eval}$ is* **contaminated** *by the dataset $D_M$ if:*

$$\exists x \in D_{eval}, \exists x' \in D_M, f(x, x') >= \tau \tag{1}$$

*where $f$ is a similarity function, and $\tau$ a similarity threshold. $\tau = 1$ corresponds to* **verbatim** *or* **strict** *contamination, but we do not restrict ourselves to this setup in this paper.*

Contamination signs often include inflated performance on evaluation benchmarks, and model memorization of entire substrings. **Contamination *detection*** refers to techniques assessing whether Equation (1) holds true or not.

**Definition 2 (Membership Inference Attack)** *A **membership inference attack (MIA)** is a function attempting to assess whether a data point was part (a member of) the training set $D_M$, and can be noted as:*

$$A : \mathcal{X} \times M(\mathcal{X}) \to \{0, 1\} \tag{2}$$

*where*

$$A(x, M(x)) = 1 \quad \text{if } x \in D_M$$
$$A(x, M(x)) = 0 \quad \text{otherwise}$$

Both concepts of contamination detection and membership inference are closely related yet fundamentally different: a membership inference attack can be used to detect contamination, but membership inference attacks can also be defined and succeed without the existence of contamination.

**Definition 3 (Model Memorization)** *Let us note $G(p)$ the model's natural language output on natural language prompt $p$. We say that $M$ has **memorized** a data point $x \in D_M$ if:*

$$\Pr[G(p_x) = x \mid x \in D_M] \gg \Pr[G(p_x) = x \mid x \sim \mathcal{D}] \tag{3}$$

*for some prompt $p_x$ that elicits $x$ (typically, $p_x$ is a truncation of $x$).*

In other words, memorization refers to the LLM's capability to recreate a training data verbatim when prompted accordingly.

## 3 *Open*-Data Contamination Detection

In the open-data setup, we have access to the (model-specific) pre-training set $D_M$. In this case, contamination detection falls back to investigating the intersection $D_M \cap D_{\text{eval}}$ for any evaluation dataset $D_{\text{eval}}$. In the following, we review existing open-data contamination detection methods, which directly compare ensembles of data points.

### 3.1 String Matching

The most straightforward method to compute overlap between textual datasets is string matching or analyzing the set of words intersecting two texts.

#### 3.1.1 Instance-Level String Matching

Several LLM pre-training reports assess contamination levels by using string matching techniques. GPT-2 (Radford et al., 2019) calculates contamination as the percentage of 8-grams from a particular evaluation set that are also found in the WebText training set. An average of 3.2% overlap between common LM datasets' test sets and the WebText dataset used for pre-training is discovered, raising the first concerns that LLM performance may be affected by memorization of pre-training data. GPT-3 (Brown et al., 2020) scans data points having a 13-gram collision with anything in the pre-training Common Crawl (C4) dataset (Raffel et al., 2020). Some earlier benchmarks sourced from Wikipedia, the Children's Book Test dataset (Hill et al., 2015), Quac (Choi et al., 2018) and SQuAD 2.0 (Rajpurkar et al., 2018), are nearly entirely contaminated. PaLM (Chowdhery et al., 2023) identifies 10 evaluation datasets at risk of contamination per their construction process, and for each dataset, partitions it into a clean and a contaminated subset based on whether at least 70% of the 8-grams of the data point can be found at least once in the training set. Both GPT-3 and PaLM authors conclude that the performance gap between clean sets and contaminated sets is mostly small across sets of various contamination degrees. GPT-4 (Achiam et al., 2023) measures cross-contamination between evaluation sets and pre-training data by computing 50-characters substring collisions. After examining GPT-4 performance gap between contaminated and clean subsets, the authors conclude that contamination overall has very little effect on the reported zero-shot results. Llama-2 (Touvron et al., 2023b) refines the methods above by using *tokens* instead of *words* to measure contamination, and asserts that a token is contaminated if it appears in any token n-gram longer than 10 tokens in both the evaluation sample and the training set. The formatting of prompts used for actual evaluation is also considered as it affects the performance more significantly. From there, the contamination level of a sample is defined as its percentage of contaminated tokens. The authors also introduce the usage of a *skip-gram* budget allowing matched spans between the evaluation sample and the training set to differ in at most four positions. Llama-3 (Dubey et al., 2024) follows a similar technique to Llama-2, but using 8-gram tokens instead of 10-gram. The ratio of contaminated tokens is varied to find the value with highest performance gain across models. Authors conclude that performance on PiQA (Bisk et al., 2020) and HellaSwag (Zellers et al., 2019) is affected by contamination. Recently, Qwen-2.5-Coder (Hui et al., 2024) removes all training data points with a 10-gram collision with the test set. Popular benchmarks HumanEval (Chen et al., 2021a), MBPP (Austin et al., 2021), GSM8K (Cobbe et al., 2021), and MATH (Hendrycks et al., 2021) are affected.

| LLM | String Matching | Contaminated Datasets |
|---|---|---|
| GPT-2 (Radford et al., 2019) | 8-gram words | CoQA (Reddy et al., 2019), LAMBADA (Paperno et al., 2016) |
| GPT-3 (Brown et al., 2020) | 13-gram words | PIQA (Bisk et al., 2020), Winograd (Levesque et al., 2012), Children's Book Test (Hill et al., 2015), WikiText-2 (Merity et al., 2016), WikiText-103 (Merity et al., 2016), enwik8, text8 |
| PaLM (Chowdhery et al., 2023) | 8-gram words | Winograd (Levesque et al., 2012), SQuAD-2 (Rajpurkar et al., 2018), WSC, ReCoRD, CB |
| GPT-4 (Achiam et al., 2023) | 50-gram characters | HumanEval (Chen et al., 2021a) |
| Llama-2 (Touvron et al., 2023b) | 10-gram tokens | HellaSwag (Zellers et al., 2019), MMLU (Hendrycks et al., 2020) |
| Llama-3 (Dubey et al., 2024) | 8-gram tokens | PiQA (Bisk et al., 2020), HellaSwag (Zellers et al., 2019), AGIEval (Zhong et al., 2023), BIG-Bench Hard (Srivastava et al., 2022), BoolQ (Clark et al., 2019), OpenBookQA (Mihaylov et al., 2018), QuaC (Choi et al., 2018), SiQA (Sap et al., 2019) |
| Qwen-2.5-Coder (Hui et al., 2024) | 10-gram words | HumanEval (Chen et al., 2021a), MBPP (Austin et al., 2021), GSM8K (Cobbe et al., 2021), MATH (Hendrycks et al., 2021) |

Table 1: Datasets flagged with a high-level of contamination for several LLM pre-training studies, assessing contamination through string matching between the pre-training corpus and the evaluation samples.

Table 1 summarizes contaminated datasets found through all string-matching contamination detection described above. Notably, PIQA (Bisk et al., 2020), Winograd (Levesque et al., 2012), HumanEval (Chen et al., 2021a) and HellaSwag (Zellers et al., 2019) are flagged by at least two contamination detection techniques.

Singh et al. (2024) review several string-matching contamination detection techniques, including the ones used by Brown et al. (2020) (Token-Match), Chowdhery et al. (2023) (Ngram-Match) and Touvron et al. (2023b) (Token-Extend). They introduce a new method called Longest-Match which measures the fraction of tokens part of the longest contaminated token span, avoiding the issue of frequent token spans artificially inflating contamination levels with other methods. Similarly to (Touvron et al., 2023b), Longest-Match also allows for skipgram. Longest-Match finds the highest Estimated Performance Gain (EPG) between contaminated and clean evaluation subsets. Through Llama-1 (Touvron et al., 2023a) and Pythia (Biderman et al., 2023) series of LLMs, the Llama pre-training corpus and The Pile (Gao et al., 2020) are found to be contaminated for many popular evaluation benchmarks, notably BigBench (Srivastava et al., 2022), HellaSwag (Zellers et al., 2019), PiQA (Bisk et al., 2020), MMLU (Hendrycks et al., 2020) and TriviaQA (Joshi et al., 2017).

### 3.1.2 Dataset-Level String Matching

Other studies analyze overlap between the training split and the evaluation or test split of the *same* dataset to assess contamination levels. While standard practice in machine learning assumes that training, validation and test splits of the same dataset are disjoint sets sampled from the same distribution, in practice, for very large datasets it may happen that the training set still overlaps with the other splits. Dodge et al. (2021) investigate the C4 (Raffel et al., 2020) dataset and study benchmark data contamination, which measures how much training or test datasets from downstream NLP tasks appear in the C4 pre-training corpus. After studying both input-label and input contamination, they discover varied contamination, ranging from less than 2% to over 50%. Li (2023b) compute the METEOR (Banerjee & Lavie, 2005) score between matched pages from CommonCrawl and queries from Bing API and consider those with scores over 0.75 as contaminated. Detected contamination levels range from 1% on Winogrande (Sakaguchi et al., 2021) to 47% on C-Eval (Huang et al., 2023). Deng et al. (2023) propose to detect contamination by retrieving top-10 documents from pre-training datasets The Pile (Gao et al., 2020) or C4 (Raffel et al., 2020), then splitting retrieved documents into 13-gram chunks, and computing overlap metrics between these chunks and the evaluation data point's chunks. TruthfulQA (Lin et al., 2021) exhibits high overlap with the pre-training datasets.

### 3.2 Embeddings Similarity

Beyond simple string matching, computing cosine similarity between embeddings offers an attractive alternative, as this process is more robust to surface-form vocabulary changes like paraphrases.

Lee et al. (2023) prevent contamination in their Open-Platypus dataset by removing test questions which have a cosine similarity (using SentenceTransformer embeddings (Reimers & Gurevych, 2019)) greater than 80% against any training item. Phi-1 (Gunasekar et al., 2023) runs a data contamination analysis between their CodeExercises dataset and the evaluation set HumanEval (Chen et al., 2021a). They show that embeddings-based retrieval between code snippets using L2 distance between normalized CodeGen-Mono 350M (Nijkamp et al., 2022) embeddings is effective, whereas n-gram overlap fails in the coding domain as it cannot capture the similarities in the logic between two coding functions.

The work from Riddell et al. (2024) provides a hybrid approach combining string matching and embeddings similarity: an *aggregated score* is designed as the maximum between the Levenshtein edit distance and the similarity from the Dolos toolkit (Maertens et al., 2022). Widely used code generation benchmarks MBPP (Austin et al., 2021) and HumanEval (Chen et al., 2021a) are contaminated with pre-training sets The Pile (Gao et al., 2020) and especially The Stack (Kocetkov et al., 2022), which significantly inflates performance of models like CodeGen-NL (Nijkamp et al., 2022).

### 3.3 Paraphrase Detection

Embeddings similarity is limited by its need to choose an appropriate similarity threshold. To improve robustness, some methods attempt to use LLMs themselves to detect contamination in LLMs, leveraging their high performance in zero-shot paraphrase detection (Witteveen & Andrews, 2019; Abaskohi et al., 2023).

Yang et al. (2023c) argue that contamination checks become challenging because of the presence of "rephrased samples", which have the same semantics but different surface form as the original sample. Therefore, they first use embeddings similarity search to get the top-k similar samples with a given test sample, then prompt a strong LLM (namely, GPT-4) to examine whether any of the top-k samples is too close to the test case. Results show that this *LLM Decontaminator* algorithm works significantly better than existing methods at tagging contaminated samples.

## 4 *Closed*-Data Contamination Detection

In closed-data contamination detection, the pre-training set $D_M$ is unknown. Since $D_M$ is not accessible, researchers have to use tools analyzing the model's behavior on the evaluation dataset $D_{\text{eval}}$.

There are two broad ways to assess closed-data contamination: examining model behavior aggregated on the whole dataset, and focusing on instance-level outputs. For the former, the main method, which we explore in Section 4.1, consists in tracking model performance on several datasets, sampled from different distributions, notably through time. For the latter, we review methods ranging from the strongest assessment of contamination to the weakest: membership-inference attacks, which assess the presence of an entire data point in the pre-training set (Section 4.2) ; then memorization of spans of text by the model (Section 4.3) ; and lastly methods analyzing model confidence on specific tokens (Section 4.4).

### 4.1 Performance Analysis

We first review a line of research assessing model contamination through simple performance analysis. $M$ is applied to different evaluation datasets, and patterns in evaluation scores may indicate contamination.

#### 4.1.1 Train-Test Performance Gap

On a dataset where the training set and the test set are sampled from the same distribution, a machine learning system is expected to perform on the test set slightly worse, or at best similar to, the training set. Zhao et al. (2024) leverage this fact to distinguish between standard fine-tuning and contamination, focusing on domain-specific setups such as financial data. They introduce a Performance Consistency Ratio (PCR), equal to the LLM performance on the test set divided by its consistency, where the latter measures the LLM's ability to produce the same output across several reasoning paths. A PCR on the training set significantly greater than on the test set is typical of fine-tuning ; the opposite case points out at contamination. Findings include potential contamination of Baichuan-13B (Yang et al., 2023a) on FinEval (Zhang et al., 2023) and of the FinMA model (Xie et al., 2023) on the FinQA dataset (Chen et al., 2021b).

#### 4.1.2 Controlled Training Setup

Jiang et al. (2024b) delve deep into the impact of evaluation data leakage within the pre-training set by running a series of controlled pre-training experiments with GPT-2 (Radford et al., 2019). They pre-train

from scratch GPT-2 in three different ways: with the original pre-training set, with additional input texts from four evaluation datasets (SST-2, MMLU (Hendrycks et al., 2020), CNN/DM (Hermann et al., 2015) and SQuAD-v1 (Rajpurkar et al., 2016)), and with input texts and labels from these same evaluation datasets. Contaminated pre-training clearly boosts performance on all downstream tasks. This was also shown by the work of Palavalli et al. (2024) in their Verbatim setup when pre-training GPT-2. Findings also reveal a U-shaped relationship between the number of times that contaminated data points are included in the pre-training, and model performance. Authors also show that the n-gram contamination detection method from Llama-2 Touvron et al. (2023b) is not effective at detecting contaminated data points, calling for more robust methods.

### 4.1.3   Performance Change Through Time

The most straightforward performance analysis consists in plotting performance against ***time***, which leverages the fact that evaluation datasets released after a LLM is pre-trained are new and should not be contaminated.

Aiyappa et al. (2023) are among the first to question the non contamination of ChatGPT training data by popular evaluation benchmarks. Their run of the January 30th, 2023 version of ChatGPT on the SemEval 2016 dataset shows a 12.5 macro-F1 score improvement compared to a prior experiment with the v1 of ChatGPT, released two months prior in November 2022. In the code domain, Roberts et al. (2023) investigate the performance of GPT-3.5 and GPT-4 on datasets released before and after the training cutoff for both models. The analysis underscores a statistically significant positive correlation for problems catalogued on GitHub prior to the GPT models' training cut-off, suggesting that their inclusion in the training data enhanced their performance on related tasks. Concurrent to their work, Li & Flanigan (2023) also plot the performance of several LLMs including GPT-3 (Brown et al., 2020) and open-source LLMs on general natural language understanding tasks against time, specifically before and after each LLM's training data cutoff date. They find that LLMs are much more likely to improve on a trivial majority baseline for datasets released before their cutoff date, suggesting some contamination. While evaluating on fresh data guarantees a contamination-free performance evaluation, it is hard to scale this method in time as it requires frequent update of evaluation sets. Besides, the work of Cao et al. (2024) shows that evaluating on data posterior to the LLM training cutoff time may not always show a decrease in performance: in the code domain, the opposite may happen, complicating the usage of training cutoffs as a contamination detection method.

### 4.2   Membership-Inference Attacks

Extracting knowledge from a language model has traditionally been addressed in the field of privacy-preserving machine learning. Adversarial attacks were crafted as **membership inference attacks (MIA)** to access training data points memorized by a black-box language model, potentially compromising sensitive private information (Carlini et al., 2021; 2022). If there is access to data points which are known not to be present in the training set, membership inference attach is referred to as **reference-based MIA** ; other cases belong to the setup of **reference-free MIA**.

### 4.2.1   Reference-based Membership-Inference Attacks

Xie et al. (2024) derive a MIA named RECALL based on the conditional language capacity of LLMs. RECALL conditions with non-member prefixes (for instance, sampled after pre-training cutoff date): log-likelihood drops more on average for member data points than for non-members, yielding a simple yet powerful MIA. The method reaches state-of-the-art ROC-AUC on the WikiMIA dataset (Shi et al., 2024). Meeus et al. (2024a) revisit MIA in the era of LLMs and define document-level MIA, which is the task of detecting whether an entire document (book, scientific paper, etc) has been trained on. A reference set is built by sampling documents known to be part of the pre-training set (e.g. Common Crawl), and data posterior to the pre-training cutoff date, which is guaranteed to be uncontaminated. Then, the authors derive features from predicted token probabilities by the LLM, which are normalized in several fashions. A meta-classifier predicts membership inference, reaching ROC-AUC as high as 0.86.

In the last stage of LLM post-training, PPO (Schulman et al., 2017) or DPO (Rafailov et al., 2023) is usually employed to train LLM on preference data. Feng et al. (2024) show that LLMs are vulnerable to MIA on the preference data, especially when using DPO. LLMs themselves can be instruction-tuned to detect if a data point is a member of their pre-training data. MIA-Tuner (Fu et al., 2024a) fine-tunes a soft prompt (Lester et al., 2021) on several target open-source LLMs to classify membership inference, and outperforms all other methods on WikiMIA. Even though at inference time, non-members are not needed, the methodology should still be assessed as a reference-based MIA due to the usage of non-members during the LLM fine-tuning.

### 4.2.2 Reference-free Membership-Inference Attacks

Fu et al. (2023) argue that both reference-based and reference-free MIA are unsatisfying with LLMs in practical scenarios. The latter rely on overfitting to assess membership inference. In the reference-free setup, they propose to prompt the target LLM to generate pseudo reference data. Semantically similar and semantically different paraphrases of target records are generated, and a membership signal is designed by measuring and comparing probabilistic variations between the target and reference models. This method reaches a ROC-AUC of 0.95 with Llama on the language modelling dataset WikiText-103. Chang et al. (2024) analyze the dynamics of next-token prediction loss for open-source Pythia and GPT-Neo LLM series. They observe that for members, the loss tends to be smoother and to decrease as context increases, whereas non-members loss curves show a flat slope as well as quite frequent outlier values. From this observation the authors derive a state-of-the-art MIA technique called CAMIA. EM-MIA (Kim et al., 2024) introduces a MIA method iteratively refining the membership score used to assess MIA and the prefix score through an expectation-maximization algorithm. The prefix score measures how discriminative the data point is in classifying members and non-members when used as a prefix. EM-MIA reaches state-of-the-art ROC-AUC scores on WikiMIA, however on a custom benchmark where members and non-members distributions are almost identical, performance drops back to random range, as for all other reported MIA methods.

### 4.2.3 Limitations of Membership-Inference Attacks

Membership-Inference attacks are hard in practice with neural networks. Duan et al. (2024) evaluate a series of MIAs on the Pythia LLMs (from 70M to 12B parameters), pre-trained on the Pile. Across types of attacks and domains, performance hardly surpasses random guessing (ROC-AUC inferior to 0.6). This failure is attributed to the common practice in LLM development of pre-training only for a single epoch and on a too large dataset, and the fact that the border between members and non-members is not clear. By splitting both groups based on time, where non-members are drawn from datasets posterior to the pre-training cutoff date, MIA accuracy drastically improves. This finding motivates the monitoring of LLMs performance through time described above. Meeus et al. (2024b) echo these findings from Duan et al. (2024): they train a trivial bag-of-words model on MIA datasets constructed by splitting on time and reach a high ROC-AUC. Authors argue that proper MIA evaluation should be done on a randomized train-test split. Concurrent work from Maini et al. (2025) also question the validity of membership inference attacks on a time-split of Wikipedia. An experiment replacing member sentences by sentences from the validation set of The Pile (Gao et al., 2020) using Pythia models (Biderman et al., 2023) shows membership inference tests still reaching an ROC-AUC of 0.7 ; proving that they merely capture a temporal distribution shift rather than actual membership to the pre-training set. Despite this lack of success of MIA on pre-training sets of LLMS, Li et al. (2024b) show that similarity between the target sample and the generated output of a LLM-powered RAG system yields a powerful MIA on the RAG database. On Natural Questions (Kwiatkowski et al., 2019) and TriviaQA (Joshi et al., 2017), the method reaches more than 0.85 in both ROC-AUC and PR-AUC.

### 4.3 Model Memorization

An intuitive approach to detect contamination in LLMs is to frame it as a **memorization** detection problem. While MIAs assess the presence of an *entire* data point into the training set, here memorization is more granular and detects memorized sequences of tokens which may not be full data points (documents), but rather subsets of them. The rate of memorized data points can be used to assess the presence of contamination. For this endeavor, it is important to first be able to quantify memorization itself.

### 4.3.1 Quantifying Memorization

Elangovan et al. (2021) highlight that some LLMs may benefit from an overlap between training and test sets, and merely memorize data points in this intersection, inflating their test set performance. This overlap is due to the common practice of simple random data shuffling in NLP tasks. By employing a bag-of-words approach to assess text similarity, the study provides a foundational method for identifying and mitigating leakage, though it acknowledges the complexity of semantic similarity measurement and the potential for more sophisticated methodologies to refine overlap detection. Magar & Schwartz (2022) find evidence of contamination across several benchmarks, impacting the performance of models like GPT-3. They propose to quantify the effect of contamination by training on a mix of general and task-specific data, then comparing performance on seen versus unseen instances to measure memorization and exploitation. This methodology highlights the nuanced relationship between memorization and exploitation. Carlini et al. (2023) propose a simple method to detect memorization by measuring the completion rates of randomly sampled prefixes from the training set, and discuss the relations between memorization with model capacity, data repetition, context size. Memorization is more widespread than previously thought, and model size, data frequency and prompt length are factors leading to an increase in memorization.

The team from Google developing the Gemma model series (Gemma Team Google DeepMind, 2024) emphasizes the potential vulnerabilities of aligned models to adversarial attacks inducing recitation of memorized training data. Through comparable methods to Anil et al. (2023), they analyze *discoverable* memorization across 10k documents from the Gemma pre-trained models' corpora. The analysis distinguishes between *exact* memorization — where the model's output precisely matches the source — and *approximate* memorization, gauged by a 10% edit distance threshold. Gemma models exhibit memorization rates comparable to those of the PaLM and PaLM-2 models.

As we have seen in Definition 3, detecting memorization sums to finding a (prefix, suffix) mapping such that the corresponding suffix can be perfectly completed when conditioning the model on the prefix. For example, Özdayi et al. (2023) append a learnable soft prompt penalized through *gradient ascent* to prevent memorizing the original suffix. Wang et al. (2024) employs a transformer-based generator to approximate such mapping of prefixes, and use it to estimate the memorization rate on unseen corpus.

### 4.3.2 Prompt Engineering

Recently, researchers have engineered elaborated prompting techniques that elicit data point completion by the LLM, where an output suspiciously close to the actual training data point indicates memorization, which in turn means contamination. Standard prompting asking the LLM to complete a data point may fail due to the LLM's alignment during RLHF (Ouyang et al., 2022b).

By prompting with an off-(training)-distribution input, Nasr et al. (2023) show that ChatGPT can regurgitate entire chunks of training data, including sensitive personal-level information. The study illustrates that even models trained with alignment techniques, aimed at reducing the emission of memorized data, can be induced to divulge substantial amounts of sensitive information, calling for the development of more robust defenses. The implications of this research are far-reaching, emphasizing the importance of addressing the privacy vulnerabilities inherent in the deployment of LLMs. Weller et al. (2023) highlight the potential for steering LLMs towards generating more factual and grounded content through effective prompting strategies. By appending instructional phrases such as *Accoding to* that encourage quoting from specific corpora, this study observes improvements in grounding as measured by the Quoted Information Precision (QUIP) Score. This method shows promise across various domains and corpora, indicating its versatility and effectiveness in leveraging LLMs' memorized knowledge for generating more accurate and reliable responses.

Golchin & Surdeanu (2023b) make use of GPT-4's abilities in *guided prompting*, an enhanced prompting process where the completion prompt includes extra information such as the dataset name. Contamination is then assessed based on the averaged difference in performance between standard and guided prompting, or if GPT-4 with in-context learning finds an exact match or two near-exact matches in the guided completions. This latter method with GPT-4 is highly accurate in identifying contamination (92%–100% accuracy). Moreover, the investigation highlights the prevalence of contamination in datasets such as AG News (Zhang et al., 2015), WNLI (Levesque et al., 2012), and XSum (Narayan et al., 2018), emphasizing the critical need

to address data integrity in LLM applications. The same authors follow up with the Data Contamination Quiz (DCQ) evaluation framework for contamination detection in black-box LLMs (Golchin & Surdeanu, 2023a). LLMs are prompted with five completion options, where one option is the ground-truth text from the original dataset, three other options are paraphrases from GPT-4, and the last choice is None. The assumption is that if the LLM picks the exact answer, it is doing so out of memorization, and the authors show that this DCQ framework finds more contamination cases than the guided prompting method. Samuel et al. (2024) also propose a quiz framework to detect contamination. Their Local Order Quiz method consists in prompting the LLM with a data point and asking it to select the subsequent data point in the original data set, among four possibilities shown in the prompt (the correct next data point, and three other random ones). A higher accuracy that the baseline 25% chance indicates that the LLM has memorized the data set. In practice, the IMDB (Maas et al., 2011) dataset shows high signs of contamination (subsequent data point selection accuracy over 80%) for all three LLMs under study: GPT-4 (Achiam et al., 2023), Claude-3 [1] and Llama-3 (Dubey et al., 2024).

### 4.3.3 Cloze Tasks

Cloze tasks constitute a special type of prompting where parts of the input is masked. Chang et al. (2023) design a cloze task in which the LLM is prompted with a book's passage, from which the character name has been replaced by a [MASK] token. Correctly predicting the character's name out of several passages per book assesses whether the LLM has seen this book during pre-training. The name cloze accuracy is very high for GPT-4 (Achiam et al., 2023) on several popular books such as *Alice's Adventures in Wonderland*, and shows a striking difference with BERT's (Devlin et al., 2019) performance on the same task. Deng et al. (2023) present the TS-Guessing method: one of the options in question-answering benchmarks test sets is hidden, and the LLM is asked to predict it. Exact match rates above 50% for OpenAI models suggest contamination on the MMLU dataset (Hendrycks et al., 2020). The work of Ranaldi et al. (2024) also uses a cloze task: they prompt the LLM to reconstruct columns names in SQL dumps, and compare the performance on an older and more recent dataset to assess contamination. GPT-3.5 accuracy falls from 33.42% to 13.21%, indicating contamination on the older Spider dataset (Yu et al., 2018).

### 4.4 Model Confidence

Going beyond simple recitation of memorized token sequences, researchers may now analyze the LLM's *confidence* to detect contamination: a (too) high confidence on some tokens may signify prior exposure to the data. Confidence analysis yields several powerful contamination detection techniques, which we highlight require gray-box to white-box access to the LLM, as researchers need to obtain the output probability distribution for each token during inference.

### 4.4.1 Likelihood

Next-token likelihood and perplexity (by extension to a whole dataset) are widely used in contamination detection.

Given a test tokens sequence $X$, the Min-K% Prob technique (Shi et al., 2024) consists in running the LLM through all tokens of $X$, and keeping track of the K% tokens with the smallest predicted probability. Then, the average between such bottom probabilities is computed and $X$ is deemed contaminated if this average is *too high*. In mathematical terms, noting $(x_1, \ldots, x_n)$ the sequence of evaluation tokens under study, Min-K% Prob is computed as :

$$\text{Min-K\% Prob}(x) = \frac{1}{|\text{Min-K\%}(x)|} \sum_{x_i \in \text{Min-K\%}(x)} \log(p(x_i|x_{<i})) \tag{4}$$

A significant drawback is the choice of value for the hyper-parameter K, which authors recommend to set to 20 as default. On WikiMIA, Min-K% Prob reaches an average ROC-AUC of 0.72, and is the highest for

---

[1] `https://www.anthropic.com/news/claude-3-family`

every LLM tested. Notably, it outperforms other methods such as perplexity or the zlib compression entropy membership-inference attack. Due to its simplicity and effectiveness, Min-K% Prob is gaining traction as one of the most popular contamination detection methods. Zhu et al. (2024) use Min-K% Prob as a first contamination detection step to identify leaked test samples before re-writing them, in order to build cleaner versions of popular evaluation datasets GSM8K (Cobbe et al., 2021) and MMLU (Hendrycks et al., 2020). Min-K% Prob performance is very high ; yet the method has a major limitation: it does not account for the token frequency distribution. LLMs put a higher probability on more frequent tokens, thus a non-contaminated sequence made of only frequent tokens might get assigned a score higher than a sequence from the training set. To fix this, DC-PDD (Zhang et al., 2024b) calibrates token probabilities and computes the cross-entropy between the token probability distribution and the token frequency distribution. The latter is estimated through a large-scale publicly available reference corpus. The tokens sequence is scored by averaging these token-level cross-entropy scores over the first occurence of each token. Contamination is assessed when this final score is above a pre-defined threshold. Min-K%++ (Zhang et al., 2024a) proposes another extension of the idea of Min-K% Prob: rather than simply getting next-token predicted probabilities, the authors propose a score which subtracts the expected log-probability and divides the score by the variance; in other words, normalizing the initial log-probability. The sequence scoring mechanism is then identical to the one of Min-K% Prob. Formally, noting the expected probability over the next token $\mu_{x<i} = \mathbb{E}_{z\sim p(.|x_{<i})}[\log(p(z|x_{<i}))]$ and its corresponding standard deviation $\sigma_{x<i} = \sqrt{\mathbb{E}_{z\sim p(.|x_{<i})}[(\log(p(z|x_{<i})) - \mu_{x<i})^2]}$:

$$\text{Min-K\%++}(x) = \frac{1}{|\text{Min-K\%}(x)|} \sum_{x_i \in \text{Min-K\%}(x)} \frac{\log(p(x_i|x_{<i})) - \mu_{x<i}}{\sigma_{x<i}} \tag{5}$$

The motivation of the score built by Min-K%++ is to assess whether an input form a mode, after observing that because of maximum likelihood training, training samples often become local maxima in the modeled distribution along each input dimension (here in the context of LLM, dimensions are tokens). Min-K%++ reaches state-of-the-art on WikiMIA, outperforming Min-K% Prob by up to 10 points.

Li (2023a) also work on token-level probabilities and compare perplexity on benchmark samples against memorized and clean baselines. The study finds significant memorization in recent models on popular reading comprehension and summarization benchmarks, while multiple-choice benchmarks show less evidence of contamination. This method provides a tool for the community to conduct rigorous contamination analysis, enabling more accurate and reliable model evaluation. Dong et al. (2024) propose two novel likelihood-based contamination detection methodologies: CDD (Contamination Detection via output Distribution) and TED (Trustworthy Evaluation via output Distribution). CDD detects data contamination by observing the peakedness in the LLM's output distribution in a black-box manner. It represents a significant improvement over existing approaches, offering average relative improvements of 21.8%-30.2% in terms of Accuracy, F1 Score, and AUC metrics. TED corrects the LLM's output distribution to mitigate the impact of data contamination on evaluation metrics, significantly reducing performance improvements attributed to data contamination across various scenarios and contamination degrees. The work of Xu et al. (2024b) proposes to use perplexity and accuracy on next n-gram prediction. They compute the relative loss in performance on these metrics on paraphrased versions of the dataset created by ChatGPT compared to the original dataset. The difference of relative loss between training and test set is later used to assess contamination or *benchmark leakage*. Experiments on GSM8K (Cobbe et al., 2021) hint at contamination for Qwen (Bai et al., 2023), Aquila and InternLM (Cai et al., 2024) series of LLMs. Zhang & Wu (2024) leverage the concept of *surprising tokens*: these are tokens where the LLM is very confident yet wrong in its prediction of the next token. The authors define a score named SURP which is the average predicted log-probability on the ground-truth token for tokens of the sequence where the model shows low entropy (and therefore, high confidence) and where this log-probability is relatively smaller. A higher SURP score is thresholded to assess contamination. The method slightly outperforms Min-K% Prob on WikiMIA.

Oren et al. (2023) take an original approach to analyze the LLM's likelihood on an evaluation dataset $D_E$. They run inference on $D_E$ and on shuffled versions of $D_E$. LLM log probabilities being statistically different on the non-shuffled dataset compared to the shuffled versions signifies contamination. The method relies on the assumption that if they are included in the pre-training set, evaluation datasets tend to be present with

| Closed-Data Method Type | Black-box | Gray-box | White-box |
|---|---|---|---|
| Performance Analysis | CAP Zhao et al. (2024) 
 Aiyappa et al. (2023) 
 Roberts et al. (2023) 
 Li & Flanigan (2023) 
 Cao et al. (2024) | | |
| Membership Inference Attacks | MaxNorm(Meeus et al., 2024a) 

 MIA-Tuner (Fu et al., 2024a) 
 SPV-MIA (Fu et al., 2023) 

 S2-MIA (Li et al., 2024b) | RECALL (Xie et al., 2024) 

 PREMIA (Feng et al., 2024) 

 CAMIA (Chang et al., 2024) 

 EM-MIA (Kim et al., 2024) | |
| Model Memorization | Carlini et al. (2023) 
 Gemma Team Google DeepMind (2024) 

 Nasr et al. (2023) 
 *According To* prompting (Weller et al., 2023) 
 DCQ (Samuel et al., 2024) 
 Local order quiz (Samuel et al., 2024) 
 Chang et al. (2023) 
 TS-Guessing (Deng et al., 2023) 
 Ranaldi et al. (2024) | Dynamic Soft Prompting Wang et al. (2024) | Magar & Schwartz (2022) |
| Model Confidence | Oren et al. (2023) | Min-K% Prob (Shi et al., 2024) 
 DC-PDD (Zhang et al., 2024b) 
 Min-K%++ (Zhang et al., 2024a) 
 Li (2023a) 
 CDD and TED (Dong et al., 2024) 
 Xu et al. (2024b) 
 SURP (Zhang & Wu, 2024) | DICE (Tu et al., 2024) 
 Liu et al. (2024) |

Table 2: Classification of closed-data contamination detection methods as per the level of access to the LLM required. Black-box methods simply need to prompt the model, while gray-box methods require access to the output tokens distributions and white-box techniques need full access to model weights.

the same default ordering. We point out that detecting the LLM's memory of a dataset canonical order is an idea also explored by Samuel et al. (2024) for a different contamination detection method.

### 4.4.2 Probing Hidden Layers

Through the use case of fine-tuning LLMs on mathematical reasoning datasets like GSM8K (Cobbe et al., 2021), Tu et al. (2024) introduce a new paradigm in (closed-data) contamination detection. Their DICE approach consists in finding the layer of the fine-tuned LLM furthest away from its counterpart layer in the not fine-tuned same LLM. Then, a MLP binary classifier is inferred on the isolated layer's weights to predict contamination. This method is particularly effective for *in-distribution* contamination detection, a setup defined by the authors as fine-tuning on a dataset from the same distribution as the evaluation dataset. In this setup, DICE reaches near-perfect ROC-AUC, outperforming all other methods, including notably Min-K% Prob.

Liu et al. (2024) also make use of the intermediate layers of a fine-tuned LLM. Their method first gathers a dataset of non-members and members ; where the former are pulled from after the pre-training cutoff or synthetically generated from ChatGPT. Then, the LLM is fine-tuned to classify membership between these two groups. Activations of intermediate layers of the fine-tuned LLM on both members and non-members are used as input of a probe classifier assessing membership. At inference time, the whole system feeds a target text to the fine-tuned LLM, extract its activations and feeds them to the probe classifier to check membership. The best layer to extract activation from is determined on a validation set. Results show greater ROC-AUC on WikiMIA (Shi et al., 2024) and ArxivMIA (Liu et al., 2024), and indicate that this probing system works better with larger models.

| Closed-Data Method Type | Contamination Detection Method | Pre-training | Instruction-tuning | RLHF |
|---|---|:---:|:---:|:---:|
| Performance Analysis | CAP Zhao et al. (2024) | ✓ | ✓ | |
| | Aiyappa et al. (2023) | | ✓ | ✓ |
| | Roberts et al. (2023) | ✓ | | |
| | Li & Flanigan (2023) | ✓ | | |
| | Cao et al. (2024) | ✓ | | |
| Membership Inference Attacks | RECALL (Xie et al., 2024) | ✓ | | |
| | MaxNorm(Meeus et al., 2024a) | ✓ | | |
| | PREMIA (Feng et al., 2024) | | | ✓ |
| | MIA-Tuner (Fu et al., 2024a) | | ✓ | |
| | SPV-MIA (Fu et al., 2023) | | ✓ | |
| | CAMIA (Chang et al., 2024) | ✓ | | |
| | EM-MIA (Kim et al., 2024) | ✓ | | |
| | S2-MIA (Li et al., 2024b) | | ✓ | ✓ |
| Model Memorization | Magar & Schwartz (2022) | ✓ | | |
| | Carlini et al. (2023) | ✓ | | |
| | Gemma Team Google DeepMind (2024) | ✓ | | |
| | Dynamic Soft Prompting Wang et al. (2024) | ✓ | | |
| | Nasr et al. (2023) | | | ✓ |
| | *According To* prompting (Weller et al., 2023) | | | ✓ |
| | DCQ (Samuel et al., 2024) | | | ✓ |
| | Local order quiz (Samuel et al., 2024) | | | ✓ |
| | Chang et al. (2023) | ✓ | | |
| | TS-Guessing (Deng et al., 2023) | ✓ | | |
| | Ranaldi et al. (2024) | ✓ | | |
| Model Confidence | Min-K% Prob (Shi et al., 2024) | ✓ | | |
| | DC-PDD (Zhang et al., 2024b) | ✓ | | |
| | Min-K%++ (Zhang et al., 2024a) | ✓ | | |
| | Li (2023a) | ✓ | | |
| | CDD and TED (Dong et al., 2024) | ✓ | | |
| | Xu et al. (2024b) | ✓ | | |
| | SURP (Zhang & Wu, 2024) | ✓ | | |
| | Oren et al. (2023) | ✓ | | |
| | DICE (Tu et al., 2024) | | ✓ | |
| | Liu et al. (2024) | | ✓ | |

Table 3: Classification of closed-data contamination detection techniques according to the model training stage(s) which they are more suitable to be applied to.

## 4.5 Practical Usage of Closed-Data Contamination Detection Methods

In the previous subsections, we reviewed the different types contamination detection methods. We now wish to give more concrete intuition in how these methods can be applied in practice.

Contamination detection techniques strongly differ in terms of model access requirement. Table 2 summarizes closed-data detection techniques organized by the minimum *level of access to the LLM* which is required. We define three levels of model access: *black-box*, which merely needs prompting access ; *gray-box*, which needs access to output tokens distributions ; and *white-box* which needs to see model weights. All methods based on performance analysis only require a black-box access, making them convenient to use. Membership-inference attacks are equally split between black-box and gray-box access requirements. Model memorization techniques mainly rely on prompting and thus only need black-box access. Lastly, for model confidence methods, which often analyze output distributions, a gray-box or even white-box model access is needed.

We also highlight that closed-data contamination detection techniques may naturally be applied to different model training stages. In Table 3, we indicate the most suitable training stage(s) where each method can be applied, among (i) pre-training, (ii) instruction-tuning and (iii) reinforcement learning from human feedback (RLHF). Prompt tuning is naturally tailored for RLHF models which can be prompted with any user query. Cloze tasks and likelihood-based methods are suitable for pre-trained models, which have captured tokens distributions. We emphasize that this categorization is flexible and not strictly defined: for instance, methods most suited for pre-trained models, like Min-K% Prob, can in theory be applied to instruction-tuned

or RLHF models. However, results should be cautiously interpreted as RLHF often smoothens overconfident predictions to align with human preferences, weakening the contamination signal.

## 5 Discussion

### 5.1 Best Practices to Avoid Contamination

Beyond detecting contaminated evaluation data points, recent works have called the community to adopt better practices to reduce contamination.

**Scanning newly released evaluation datasets** Sainz et al. (2023) argue to develop automatic or semi-automatic approaches to detect contamination for new benchmarks and design mechanisms to flag those works with contamination. This process goes hand in hand with creating new, non-contaminated evaluation datasets.

**Evaluating on a wide spectrum** To comprehensively and fairly evaluate LLMs, Zhou et al. (2023) suggest to use more benchmarks from diverse sources, covering both basic and advanced capabilities. Furthermore, they recommend employing multiple task prompts for benchmark tests to derive model performance that is more stable and reliable. Additionally, they highlight the necessity of conducting data decontamination checks between pre-training data and any evaluation data when using benchmarks.

**Encrypting evaluation datasets** Jacovi et al. (2023) advocate for the encryption of publicly released test data using a public key, coupled with licensing agreements that prohibit derivative distribution. They recommend implementing training exclusion controls for closed APIs and safeguarding test data by refusing to perform evaluations without such controls. Moreover, they suggest avoiding data that appears with the solution and releasing internet-derived data along with the corresponding web-page context.

**Not leaking data to closed-source APIs** Aside from technical methodologies, Balloccu et al. (2024) conduct a systematic literature review of 225 papers and carefully detail data leakage from them to closed-source models like the GPT-series. Overall, they conclude that ~42% of reviewed papers leaked data to GPT-3.5 and GPT-4 for a total of ~4.7M benchmark samples across 263 benchmarks. The authors report evaluation malpractices and propose a list of suggested practices for evaluating closed-source LLMs.

Several data contamination mitigation techniques are detailed in the survey work of Xu et al. (2024a).

### 5.2 Towards Open-Source Pre-Training Data

As we have seen in this study, access to the pre-training data is the first fundamental characterization of contamination detection technique. When the pre-training data are publicly available, the task of defining membership to the pre-training set becomes drastically simpler. To facilitate contamination detection, it is important for the research community to move towards the general usage of open-source pre-training sets.

Duan et al. (2024) introduce the **MIMIR** dataset, constructed from The Pile training set (Gao et al., 2020), on the which popular open-source models such as the Pythia (Biderman et al., 2023) and GPT-Neo[2] model series are pre-trained. MIMIR covers domain-specific subsets (e.g. GitHub, Wikipedia, Arxiv) and extracts members through several n-gram sizes, allowing researchers to work on multiple levels of membership granularity. MIMIR has become widely used in MIA research with LLMs. **OLMoMIA** (Kim et al., 2024) introduces a membership inference dataset centered around the open-source OLMo-7B LLM (Groeneveld et al., 2024). Knowing that a pre-training epoch roughly consists of 450k, training data seen before 100k pre-training steps is considered as member, and training data 400k and 450k steps is non-member. **Dolma-Book** (Zhang & Wu, 2024) also leverages the publicly available pre-training data from OLMo, and samples non-member books from books from the Project Gutenberg[3] dated after January 1st, 2024.

---

[2] https://github.com/EleutherAI/gpt-neo
[3] https://www.gutenberg.org/

### 5.3 New Evaluation Benchmarks

Researchers have crafted new datasets specifically introduced with the aim to provide contamination-free LLM evaluation.

**Benchmark Perturbation**   Several new datasets are crafted by *perturbing* existing benchmarks, for instance through rephrasing instructions or adding more complex instructions. Perturbed data points are more challenging for LLMs, and can be designed to be adversarial examples aiming to fool the LLMs. Moreover, Alzahrani et al. (2024) point out that slight perturbations to existing benchmarks may disrupt model rankings on leaderboards, calling for better evaluation benchmarks. Such benchmarks include:

- **GSM-Plus** (Li et al., 2024a). Using the 1,319 questions of the grade-school level maths word problems GSM8K dataset (Cobbe et al., 2021) as seed dataset, the authors apply eight perturbations of five different types: numerical variation, arithmetic variation, problem understanding, distractor insertion and critical thinking. The dataset is generated with GPT-4 (manually verified by humans) and has 10,552 questions in total. High relative performance drop rates of above 30% are observed for open-source models. Several elaborate prompting techniques such as chain-of-thought fall short of providing meaningful improvement, but a custom compositional prompting technique provides notable performance improvement.
- **MATH-Perturb** (Huang et al., 2023) introduces two flavors of variations to 279 problems from the hardest level (level 5) of MATH dataset (Hendrycks et al., 2021): slightly different problems solvable with the same method (MATH-P-Simple), and problems needing a different approach (MATH-P-HARD). Performance degrades notably for all LLMS on MATH-P-HARD. Interestingly, a lot of errors can be traced to memorization: the model simply blindly applies the same technique used to solve the original problem.

**Dynamic Evaluation**   Some new datasets discard the *static* paradigm of existing LLM evaluation datasets, and instead propose a *dynamic*, ever-changing or periodically updated benchmark to avoid contamination.

- **LatestEval** (Li et al., 2023b) leverages the most recent texts to create dynamic reading comprehension evaluations. The benchmark is constructed in three steps: (*i*) collect the latest texts; (*ii*) extract key information, and (*iii*) construct questions based on the extracted information through template-filling or LLMs.
- **KIEval** (Yu et al., 2024) is an interactive evaluation framework incorporating an LLM-powered "interactor", which can ask follow-up questions in multi-round dialogue that leads to contamination-resilient evaluation.
- **LiveCodeBench** (Jain et al., 2024) continuously collects new problems from LeetCode, AtCoder and CodeForces to provide a contamination-free code generation benchmark.
- **LiveBench** (White et al., 2024) is a challenging benchmark made of 18 tasks from 6 categories: math, coding, reasoning, language comprehension, instruction following, and data analysis. Tasks are more complicated versions of existing benchmarks, such as BigBench-Hard (Srivastava et al., 2022), and are evaluated against an objective ground-truth rather than with human or LLM preference. To avoid contamination, one sixth of questions are replaced every month, chosen among oldest and easiest tasks.
- **Chatbot Arena** (Zheng et al., 2023) is a popular crowdsourced LLM evaluation platform. Humans submit queries and are shown answers from two LLMs which quality they have to rank. By construction, this benchmark is not set in stone and changes LLM rankings as more users interact with the platform. Besides, the design of Chatbot Arena limits contamination as user-generated prompts and model response rankings are less likely to be trained on.

Several dynamic benchmarks are specifically crafted for membership-inference attacks:

- **WikiMIA** (Shi et al., 2024) is a dynamic benchmark made of Wikipedia events created after model training (*i.e.,* after 2023). ChatGPT is leveraged to paraphrase examples for evaluation, as also performed in CleanEval (Zhu et al., 2023).

- **WikiMIA-24** Fu et al. (2024a) tackle WikiMIA's as well as any publicly released dataset issue: cutoff dates will need to be adjusted as time goes on. To address the issue, the authors extend WikiMIA's cutoff to March 2024.

- **ArxivMIA** (Liu et al., 2024) gathers paper abstracts from the Computer Science and Maths sections of Arxiv. Abstracts dated after 2024 are considered non-members of LLMs pre-training sets.

- **PatentMIA** (Zhang et al., 2024b) constructs a benchmarks where non-members are 5,000 patents from Google-Patents which are in Chinese language and dated after March 1st, 2024, and members are 5,000 such patents dated from before January 1st, 2023.

**Protected Evaluation**  Another strategy to avoid contamination of evaluation data is to protect it by locking it out of the public domain. **Termite** (Ranaldi et al., 2024) is a new text-to-SQL dataset, locked out of public access through search engines via an encryption key. Termite is made of handcrafted databases, each paired with around five queries, designed to match properties of the Spider dataset (Yu et al., 2018), which shows high signs of contamination on GPT-3.5.

Scaling datasets used for contamination detection is critical, and such datasets should follow advancements in scale and complexity of datasets used to evaluate LLMs. Samuel et al. (2024) emphasize that as of late 2024, there remains a wide discrepancy between datasets used to evaluate LLM performance and datasets used to detect contamination, with the latter usually lagging the former by several years.

## 5.4 Future Challenges

Given the fast and ever-changing landscape of machine learning research, future directions for contamination detection in LLMs could encompass a broad variety of methodologies and technological aspects. We highlight a few potentially critical areas of focus:

**Real-time contamination detection systems**  Real-time data contamination detection systems that can monitor data streams continuously and alert users to potential contamination events as they happen would be of upmost importance in critical applications like finance (Frésard et al., 2011) and healthcare (Gao et al., 2022) where data integrity is paramount and model reliability critical. Given the sheer volume of new data uploaded daily on the Internet, solving this task requires major technological breakthroughs. The key difficulty lies in achieving high recall, to ensure that no contaminated data sample slips through undetected. We hypothesize that compression (Han et al., 2015), distillation (Hinton et al., 2015; Sanh et al., 2019) and quantization (Banner et al., 2018; Sun et al., 2020) techniques could play a key role in solving this problem, by decreasing model size, accelerating inference and scaling edge-device deployment of anomaly detection models or closed-data contamination detection techniques. Besides, leveraging federated learning and edge computing could enable decentralized, privacy-preserving detection at the source of data generation.

**Bypassing contamination detection**  Dekoninck et al. (2024) show a very effective way to bypass several existing contamination detection methods: Evasive Augmentation Learning (EAL) consists in paraphrasing benchmarks with GPT-4 and fine-tuning the LLM on the paraphrased data. This method can bypass even oracle-access contamination detection methods, calling for the development of more robust methods. An intuitive future line of defense to such techniques is to incorporate large-scale adversarial training (Jia & Liang, 2017; Liu et al., 2020) into contamination detection models, and train these models on benchmarks which have been paraphrased or noised. We expect to witness a *cat-and-mouse* game between techniques such as EAL, which contaminate under the radar, and contamination detection techniques catching up to the finest transformations of benchmarks, such as paraphrases.

**Ethical and legal data frameworks**  Ethical and legal frameworks (Chang, 2021) are needed to govern the collection, usage, and management of LLM pre-training data. A strong legal framework will help prevent the inclusion of data from unethical sources or from copyrighted sources, as well as help prevent the contamination of evaluation benchmarks into widely used pre-training data sources (e.g., CommonCrawl). However, adopting such a worldwide pre-training data legal framework would be challenging due to several reasons. First, the lines are blurry between public and private data, and many web-crawled datasets include a mix of

public, open-access and proprietary data. Jurisdiction surrounding data varies from a country to another, complicating the adoption of global standards. Besides, the authorization to train machine learning models on copyrighted data remains unclear in many jurisdictions to this day. Another major blocker is the fact that pre-training data is nowadays treated with secrecy by large technological companies releasing flagship LLMs, as publishing the exact pre-training data would enable their competitors to catch up with their own LLMs. Lastly, enforcing and auditing a legal framework surrounding pre-training pipelines in practice would be very complex and costly. Above all, legal frameworks on pre-training data are needed to protect personal data privacy. The General Data Protection Regulation (GDPR), acted in 2018 in the European Union (EU), imposes strict rules on how organizations can collect, store and use personal data from EU citizens. With the exponential growth of pre-training corpora, it becomes harder and harder yet critical to ensure that personal data does not get included into LLM training sets. Once an LLM has seen and memorized personal data, there is no clear mechanism for an individual to exercise their right to erase their personal data from the LLM's knowledge. In the near future, developing contamination detection techniques that do not compromise individual privacy will be crucial (Lin et al., 2010; Hayes & Ohrimenko, 2018).

## 6 Conclusion

As LLMs rapidly evolve and the training data become more extensive, LLMs' performances are inevitably biased by contamination. Eventually, most (if not all) target-answer-based public evaluation datasets end up in commonly used pre-training data dumps, which are updated every few months. Therefore, to help detect and quantify contamination, in this survey, we investigated, structured, and classified the current landscape of contamination detection in LLMs. Depending on a researcher's level of access to the data and to the LLM parameters, our survey can quickly guide them toward a set of relevant tools to detect contamination.

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
