# OpenReview forum: "A Comprehensive Survey of Contamination Detection Methods in Large Language Models"
_TMLR — Accepted by TMLR_

### Review · Reviewer_67L4 · 2025-04-27

**Summary Of Contributions:**

This survey focuses on the data contamination issue of LLMs. It comprehensively reviewed the contamination detection methods for both open-data and closed-data settings. The work also discussed some practices on how to avoid data contamination and build contamination-free benchmarks.

**Audience:**

Yes

**Broader Impact Concerns:**

Data contamination and de-contamination is recognized as important for reliable LLM evaluation. Thus there are considerable social impacts and broader concerns about this issue. It would be nice if the authors provided additional discussions about this aspect.

**Claims And Evidence:**

No

**Requested Changes:**

Please see above weaknesses and questions. Overall, more appropriate organization and deeper analysis are required.

typos
- page 2, "enabling enabling"

**Strengths And Weaknesses:**

Strengths
- The authors provide a comprehensive overview covering over 100 papers in this topic. They also give a structured organization for data contamination. This would help researchers to get into this area.
- The topic selected, data contamination, is of great importance for the healthy development of LLMs

Weaknesses
- Definition of contamination and other terms
	- What is the formal definition of data contamination? Are there any established definitions in the context of LLMs or other fields? If not, can the authors discuss this issue, especially data contamination.
	- As for the proposed framework, there also lacks clear definition and proper demonstration on some lines of work. For instance, what is membership inference attack and how is it different from model memorization methods? How MIA is used in data contamination detection? Such basic questions require more demonstration.
- As an overall survey of this field, it is important to provide an all-round look at the current status, which is missing in the manuscript. Specifically, what is the cause of data contamination? What are the potential harms and broader impact of contamination? Are there any reports or well-known conclusions about contamination of LLMs? Are there any real-world applications or services for data contamination detection?
- Despite the overview of contamination detection of LLMs, the discussion on academic and industrial efforts on how to avoid data contamination. I feel the current version could be further improved. For example, the Dynamic Evaluation section mostly involves MIA benchmarks only, which have limited connections with data contamination. There are more dynamic benchmarks than LiveCodeBench, such as Chatbot Arena, LiveBench. Moreover, some benchmarks adopt perturbation-based manipulation of the original dataset to handle potential contamination, such as GSM-Plus, Math-Perturb etc. These works should also be included and discussed.
- Some discussions feel lack of depth and incomplete.
	- For the different kinds of data contamination detection, are they suitable for all kinds of contamination? What are the specific settings of contamination detection tasks? Are there any differences between contamination in pre-training, finetuning, and RLHF?
	- In Section 4.2, the Open-Source Pre-training Data paragraph, the authors mainly introduced three pre-training datasets, but did not go deep on the main contributions and differences of the datasets. How do the datasets benefit the data contamination research? What are their main conclusions? It seems that they are MIA datasets and the relevance is not well-elaborated.

References

[1] Chatbot Arena: An Open Platform for Evaluating LLMs by Human Preference

[2] LiveBench: A Challenging, Contamination-Limited LLM Benchmark

[3] Gsm-plus: A comprehensive benchmark for evaluating the robustness of llms as mathematical problem solvers

[4] MATH-Perturb: Benchmarking LLMs' Math Reasoning Abilities against Hard Perturbations

---

> ### Author Response · Authors · 2025-05-13
> **Addressing weaknesses**
>
> We thank the reviewer for their comments.
> Here are the changes we have made to address all their concerns:
> - Regarding the formal definition of terms such as contamination or membership inference attacks, we have added a dedicated section, Section 2, defining all terms. We believe that this greatly strengthens the paper.
> - Potential harms, and real-world applications were already mentioned in the Introduction.
> We have added a paragraph (paragraph #2) in the Introduction covering potential causes.
> - We have added Chatbot-Arena and LiveBench to the dynamic benchmarks paragraph in subsection 5.3 ; as well as created a new paragraph covering perturbation-based benchmarks including GSM-PLUS and MATH-PERTURB.
> - We have re-structured section 4 on closed-data methods such that the final subsection 4.5 focuses on practical usages of detection methods. It contains Table 2 on access-level, as well as a new Table 3 which we have added and which indicates the most suitable training stage for each detection method.
> - We have re-structured the Open-Source Training Data paragraph and expanded it, it is now sub-section 5.2

---

### Review · Reviewer_VDsX · 2025-05-06

**Summary Of Contributions:**

The manuscript is a survey, and it covers contamination detection methods for two settings Open-Data Contamination Detection and Closed-Data Contamination Detection.

**Audience:**

Yes

**Broader Impact Concerns:**

I think the authors should have a stand alone broader impact section by including discussions from the section of "Ethical and legal data frameworks."

**Claims And Evidence:**

Yes

**Requested Changes:**

Incorporating the discussion from the weaknesses section would strengthen the paper and reinforce my recommendation for acceptance.

**Strengths And Weaknesses:**

Strength:
The paper summarizes the contamination detection methods for open-data contamination detection and closed-data contamination detection. The coverage seems comprehensive.

Weakness:
1. The section regarding Quantifying Memorization seems to lack additional details. For example, several papers regarding discoverable memorization are not discussed [1, 2, 3].
2. In the section of best practices to avoid contamination, the authors did not discuss evaluation settings like chatbot arena [4]. This kind of evaluation will no doubt avoid the problem of dataset contamination. It may also have some weakness, for example the model can be specifically tuned to align the preference of humans to boost its scores. The authors should discuss this approach or similar approaches in the paper.

[1] Quantifying Memorization Across Neural Language Models, ICLR 2023.

[2] Controlling the extraction of memorized data from large language models via prompt-tuning, ACL 2023.

[3] Unlocking Memorization in Large Language Models with Dynamic Soft Prompting, EMNLP 2024.

[4] Chatbot Arena: An Open Platform for Evaluating LLMs by Human Preference

---

> ### Author Response · Authors · 2025-05-13
> **Incorporated the papers mentioned by the reviewer**
>
> We thank the reviewer for their comment.
> We have added papers [1,2,3] to subsection 4.3, which indeed makes it more comprehensive.
> We have also added the Chatbot-Arena [4] to section 5.3

---

> > ### Comment · Reviewer_VDsX · 2025-05-21
> > **Response**
> >
> > I appreciate the authors' efforts in revising the paper. I think they did a good job for addressing my concerns.

---

### Review · Reviewer_fdZF · 2025-05-06

**Summary Of Contributions:**

Data contamination refers to the phenomenon where improper data (i.e., test data leakage or unethical data) has been mixed into model training data. This survey defines contamination as “any leakage of the evaluation data within the training data”, and conducts a systematic literature review on current data contamination detection techniques. The authors organize existing works through a taxonomy distinguishing between open-data scenarios (where model pre-training data is known) and closed-data scenarios (where pre-training data remains unknown). The review encompasses over 100 papers up to early 2025. Additionally, the authors provide practical recommendations for dataset construction and release to prevent contamination, along with discussions of future challenges.

**Audience:**

Yes

**Broader Impact Concerns:**

Since this is a survey paper, there may not be any ethical concerns to address.
However, the authors could include a Broader Impact section discussing the positive impact that the contamination detection techniques may have.

**Claims And Evidence:**

No

**Requested Changes:**

1. Please consider supplementing discussions of missing survey works ([1], [2]) with analysis of their distinct contributions.
2. A detailed methodology section documenting literature search protocols and selection criteria should be added.
3. I recommend inserting a formal task definition section after the introduction part, specifying mathematical formulations and evaluation frameworks for contamination detection. This would enhance conceptual clarity for readers.
4. Technical presentation issues (hyperlink errors, typographical consistency in mathematical notation) require correction.

**Strengths And Weaknesses:**

**Strengths**
1. This survey paper demonstrates timeliness by including the most recent works. The proposed taxonomy based on data accessibility provides clear coarse-grained categorization.
2. The authors provide insights that go beyond a mere systematic summary of existing works. The discussion on contamination mitigation strategies and detection challenges is valuable for the research community.

**Weaknesses**
1. Based on my literature research, several relevant survey works on data contamination appear missing from the discussion: [1] (accepted in 2024), [2] (2025 preprint). Furthermore, the designation of Xu et al. (2024a) [3] as “concurrent work” raises questions given its publication date. While I am not familiar with the review iterations of this paper in TMLR, the timing of this version suggests that classifying [3] as “concurrent work” may be inappropriate.

2. The literature search and selection methodology lacks documentation. Critical details about selection criteria, sources, and search strategies remain unspecified, which are essential components for a survey paper.

3. Although the introduction provides background about data contamination , the paper would benefit from an extra section formally defining the contamination detection task and establishing quantitative evaluation metrics prior to discussing current techniques.
4. Presentation issues require attention:

- Page 8: Hyperlink misalignment. Current hyperlink of literature incorporates the next page header.

- Page 9: Inconsistent notation (italicized $K$ vs. regular K) in Min-K detection methodology.

---

**References**
- [1] Deng, C., Zhao, Y., Heng, Y., Li, Y., Cao, J., Tang, X., & Cohan, A. (2024). Unveiling the Spectrum of Data Contamination in Language Model: A Survey from Detection to Remediation. In Findings of the Association for Computational Linguistics ACL 2024 (pp. 16078-16092).
- [2] Cheng, Y., Chang, Y., & Wu, Y. (2025). A Survey on Data Contamination for Large Language Models. arXiv preprint arXiv:2502.14425.
- [3] Xu, C., Guan, S., Greene, D., & Kechadi, M. (2024). Benchmark data contamination of large language models: A survey. arXiv preprint arXiv:2406.04244.

---

> ### Author Response · Authors · 2025-05-13
> **Incorporating changes**
>
> We thank the reviewer for their comments.
> Here are our actions taken to incorporate all requested changes:
> (1) We highlight that we are already citing [1] at the end of the Introduction.
> (2) We have added a sentence in the Introduction explaining our papers selection process.
> (3) This is a great suggestion and we have added a Section 2 formally defining contamination, membership inference attacks and memorization.
> (4) We have fixed both typos.

---

### Author Response · Authors · 2025-05-13
**Thank you for your reviews**

We warmly thank all reviewers for their constructive reviews. We sincerely believe that all reviews were of great quality, and have done of our best to incorporate the changes which the reviews entail into a new version of the paper. As a result of these changes, the draft is now stronger.
Several reviewers have mentioned a lack of formal definitions, which is now handled by the new Section 2. All reviewers have mentioned missing papers from our literature, which we have added. The new draft also contains heavy modifications to the Introduction (section 1), the end of the section on closed-data detection methods (section 4), and the Discussion (section 5). Table 3 has been added to 4.5 to showcase the most suitable training stages for each closed-data method.
All edits to the draft are shown in red.

---

### Author Response · Authors · 2025-06-09
**Following up on our response to reviews.**

We kindly remind reviewers who have not done so yet to give a feedback to our response to their initial reviews.

---

### Decision · Action_Editor_epmp · 2025-06-16

**Recommendation:** Accept with minor revision

**Additional Comments:**

This paper was borderline, as surveys on LLM contamination, covering both the phenomenon itself and its detection, already exist. As a result, it was unclear to me and several other reviewers what distinguishes this survey from previous ones, beyond offering a more up-to-date and thorough literature review and a different framing perspective. If there are other distinguishing features, I strongly encourage the authors to clarify them before the final version.

In addition, I agree with Reviewer *fdZF* that labeling some existing works as "concurrent" is inappropriate, and I ask the authors to revise their wording accordingly.

That said, most reviewers, including myself, recognize that the paper makes a valuable contribution to the literature. Its strengths, particularly the detailed taxonomy and synthesis, are sufficient to warrant acceptance.

Nonetheless, I would like to pass along a few suggestions from one of the reviewers that I believe would further improve the paper:

* Please clearly state what distinguishes this survey from prior ones.
* The discussion of long-term developments in this field could be deepened. For example:

  * In the “Future Challenges” section, only three critical areas are briefly listed. How might we proactively prepare for these challenges?
  * Regarding the ethical/legal frameworks: What are the ideal or desired outcomes? What concrete steps or collaborative efforts should be undertaken by various stakeholders?

I encourage the authors to consider these suggestions prior to final acceptance.

**Audience:**

Yes

**Audience Explanation:**

Most reviewers and I agree that the paper addresses a timely topic, making this work highly relevant to the community.

**Claims And Evidence:**

Yes

**Claims Explanation:**

The claims made in the submission are generally supported by clear and well-structured evidence, particularly in the presentation of an organized taxonomy and the review of over 50 detection techniques. The distinction between open-data and closed-data contamination use cases is clearly articulated and helps frame the landscape effectively. The categorization into finer-grained detection techniques is thorough and demonstrates familiarity with the technical assumptions of different approaches.

---

> ### Author Response · Authors · 2025-07-08
> **Made the required changes**
>
> Dear Action Editor,
>
> We warmly thank you for you decision.
>
> We have submitted a Camera-ready version of the paper, which includes the changes that you demanded:
> - In the Introduction, we re-structured the section and included a subsection comparing our survey to each of the other surveys on contamination. We also stopped claiming concurrency with these other surveys.
> - In the Discussion, we expanded subsection 5.4 (especially the legal framework paragraph), and included future avenues of research.